# Spectral Representation via Data-Guided Sparsity for Hyperspectral Image Super-Resolution [note 1]

**DOI:** 10.3390/s19245401

**Published:** 2019-12-07

**Authors:** Xian-Hua Han, Yongqing Sun, Jian Wang, Boxin Shi, Yinqiang Zheng, Yen-Wei Chen

**Affiliations:** 1Graduate School of Science and Technology for Innovation, Yamaguchi University, 1677-1 Yoshida, Yamaguchi 753-8511, Japan; 2Media Intelligence Lab, NTT Corporation, Kanagawa 239-0847, Japan; yongqing.sun.fb@hco.ntt.co.jp; 3College of Information Science and Engineering, Ritsumeikan University, 1-1-1 Nojihigashi, Kusatsu 525-8577, Japan; jwang@sdnu.edu.cn (J.W.); chen@is.ritsumei.ac.jp (Y.-W.C.); 4School of Electronics Engineering and Computer Science, Peking University, Beijing 100871, China; shiboxin@pku.edu.cn; 5National Institute of Informatics, Tokyo 101-8430, Japan; yqzheng@nii.ac.jp

**Keywords:** hyperspectral image superresolution, sparse representation, data guided sparsity, spectral mixing, local content similarity

## Abstract

Hyperspectral imaging is capable of acquiring the rich spectral information of scenes and has great potential for understanding the characteristics of different materials in many applications ranging from remote sensing to medical imaging. However, due to hardware limitations, the existed hyper-/multi-spectral imaging devices usually cannot obtain high spatial resolution. This study aims to generate a high resolution hyperspectral image according to the available low resolution hyperspectral and high resolution RGB images. We propose a novel hyperspectral image superresolution method via non-negative sparse representation of reflectance spectra with a data guided sparsity constraint. The proposed method firstly learns the hyperspectral dictionary from the low resolution hyperspectral image and then transforms it into the RGB one with the camera response function, which is decided by the physical property of the RGB imaging camera. Given the RGB vector and the RGB dictionary, the sparse representation of each pixel in the high resolution image is calculated with the guidance of a sparsity map, which measures pixel material purity. The sparsity map is generated by analyzing the local content similarity of a focused pixel in the available high resolution RGB image and quantifying the spectral mixing degree motivated by the fact that the pixel spectrum of a pure material should have sparse representation of the spectral dictionary. Since the proposed method adaptively adjusts the sparsity in the spectral representation based on the local content of the available high resolution RGB image, it can produce more robust spectral representation for recovering the target high resolution hyperspectral image. Comprehensive experiments on two public hyperspectral datasets and three real remote sensing images validate that the proposed method achieves promising performances compared to the existing state-of-the-art methods.

## 1. Introduction

Hyperspectral (HS) imaging is an emerging technique for simultaneously obtaining a set of images of the same scene on a large number of narrow band wavelengths. The rich spectral information significantly benefits for analyzing the characterization of the obtained scene and greatly enhances the performance in different computer vision tasks including object recognition and classification, tracking, and segmentation [1,2,3,4,5]. In addition, HS imaging has also been a promising tool for different applications in disease diagnosis in medical images [6], land resource management/planning in remote sensing [7,8], etc. Although HS imaging can provide high spectral resolution, it imposes a severe limitation on the spatial resolution compared with the general RGB cameras. In order to guarantee a sufficient signal-to-noise ratio, enough exposure amount is needed for each narrow wavelength window, which can be generally solved in the existing hyperspectral cameras via collecting exposures in a much larger spatial region than the common RGB cameras, resulting in much lower spatial resolution. The low spatial resolution may result in the possible spectral mixture of different materials and thus restricts its performances for scene analysis and understanding. On the other hand, the high spatial resolution multi-spectral (e.g., RGB and RGB-NIR) images are easily available for the same scene with the common color cameras. Therefore, how to fuse the available low resolution HS (LR-HS) image and the HR-RGB image to generate a high resolution HS (HR-HS) image has become an attractive topic. The fusion method can effectively utilize the spectral correlation property in the LR-HS image and the detailed spatial structure in the HR-RGB image and thus generate a more accurate HR-HS image.

This study aims to generate an HR-HS image according to the spectral mixture model with a sparse constraint. Formally, in the spectral mixture model, a pixel spectrum z in an HS image can be assumed to be a weighted combination of *K* spectral bases ${\left\{b_{k}\right\}}_{k=1}^{K}\in R^{+}$. Each spectral base is called an endmember, which denotes the pure spectrum of a signal material such as “water” or “grass” in the physical meaning or a spectral prototype (atom) in a mathematical formula. Specifically, to ensure the non-negativity of the composite percentage in physical meaning, the pixel spectrum z is generally approximated by a nonnegative linear combination as:(1)$$z=\sum_{k}^{K}b_{k}\alpha_{k}$$
where $\alpha_{k}$ is the composite percentage (weighted coefficient) of the *k*th spectral base. Generally, both spectral bases ${\left\{b_{k}\right\}}_{k=1}^{K}$ and the weighted coefficient $\alpha_{k}$ are unknown and thus result in an NP-hard problem. In order to make the problem solvable, prior knowledge is required to restrict the solution space to constrain the solution toward a robust representation. Various constraints on the composite weights and the spectral bases have been exploited, where the sparse constraints [9,10,11,12,13], via using the $l_{0}$ norm: ${\left\|\alpha \right\|}_{0}$ or the $l_{1}$ norm: ${\left\|\alpha \right\|}_{1}=\sum |\alpha_{k}|$ on the composite weights, are the most popular ones. Although the actively investigated sparse spectral representation has manifested impressive performance, the conventional methods [14,15,16,17] usually exploit an identical strength of the sparse constraint on all the spectral samples. The identical sparse constraint does not consider the individual property of each spectral sample, which may not meet the practical situation.

This study proposes a novel spectral representation with the data guided sparse constraint for the HS image superresolution task. Motivated by the fact that the mixing degree of each pixel spectrum should be different from each other, we investigate the possible purity (mixing degree) of each pixel in the HR-RGB image. Intuitively, the spectrum of a pixel that has a similar spectrum as the surrounding pixels should be pure, otherwise it would lie on an edge between different materials. Then, we explore the local content similarity of a focused pixel as an indication of spectral purity and incorporate the material purity as the sparsity constraint into the non-negative sparse spectral representation. Our proposed method adaptively adjusts the strength of the sparsity constraint in the spectral representation based on the explored material purity and thus can produce more robust spectral representation for recovering the target HR-HS image. To our best knowledge, this is the first time the spectral purity has been adopted to adjust the sparsity strength of the representation coefficients for the HS image superresolution. Compared with our previous work in [18], we further conduct an ablation study to compare our proposed method, the baseline sparse representation, and the generalized simultaneous orthogonal matching pursuit (G-SOMP+) [13] to optimize under different parameters. We also provide the comprehensive results’ comparison in both performance and computation cost. In addition, we extend our method for the superresolution of three real satellite images with many more bands: 103, 128, and 224, respectively, compared with the HS image datasets used in [18]. Experimental results on two public HS datasets and the real satellite images manifest that our propose method achieves impressive performances compared with the existing state-of-the-art methods.

## 2. Related Work

Although the HR-HS image has proven potential prospects in different application fields ranging from remote sensing to medical imaging, it is still difficult to achieve high resolution simultaneously in both spatial and spectral domains due to hardware limitations [19]. On the other hand, the HR-RGB images are easily obtained with a common color camera. Thus, this has inspired much research effort to generate the HR-HS images via image processing and machine learning techniques based on the available LR-HS and HR-RGB images. In the remote sensing field, an HR panchromatic image is usually available accompanying the LR multi-spectral or HS image, and fusing these two images to generate an HR-MS or HS image is generally known as the pansharpening technique [20,21,22,23,24,25]. In this scenario, most approaches perform reliable illumination restoration based on intensity substitution and projection with the explored hue saturation and principle component analysis [20,21], which usually result in spectral distortion in the generated HR image [26].

Recently, HS image superresolution based on matrix factorization and spectral unmixing has been actively investigated [9,10,27,28]. Spectral matrix factorization and unmixing based methods assume that the HS observations can be decomposed into two matrices, where one represents a set of bases as the spectral reflectance functions (the spectral response of the pure material) and the other is the corresponding coefficient matrix denoting the composite fraction of each material on each pixel location. Yokoya et al. [28] investigated a coupled non-negative matrix factorization (CNMF) to generate an HR-HS image from a pair of HR-MS and LR-HS images. Although the CNMF approach achieved promising spectral recovery performance, its solution was generally not unique [29]. Thus, the spectral recovery results were not always satisfactory. Lanaras et al. [10] proposed a coupled spectral unmixing strategy for HS superresolution and estimated the HR-HS image via simultaneously minimizing the reconstruction errors of the observed LR-HS and HR-RGB images using the proximal alternation linearized minimization method for optimization. Considering the physical meaning of the reflectance functions and the implementation robustness, the number of pure materials in the observed scene is often assumed to be smaller than the spectral band number, which does not always meet the real application.

There have existed many image restoration methods [14,15,30,31] for nature images. Motivated by the successes of sparse representation in natural image analysis, the sparsity promoting approaches have been widely applied for HS superresolution [11,12,13]. The sparse representation based method does not need to impose explicitly the physical meaning constraint on the bases and thus permits over-complete bases. Grohnfeldt et al. [11] proposed a joint sparse representation of the corresponding HS and MS (RGB) patches. They firstly learned the joint HS and MS (RGB) patch dictionaries using the prepared corresponding pairs and estimated the sparse coefficients of the combined MS and previously reconstructed HS patches for each individual band. This method mainly focused on reconstructing the local structure (patch) and completely ignored the correlation between channels. Therefore, several works [12,13] explored the sparse spectral representation instead of the local spatial structure. Akhtar et al. [12] explored a Bayesian dictionary learning and sparse coding algorithm for HS image superresolution and manifested impressive performance. Dong et al. [32] investigated a non-negative structured sparse representation (NSSR) approach to recover an HR-HS image, which imposed the similarity constraints in a spatial structure region to estimate robust sparse representation and proposed to use the alternative direction multiplier method (ADMM) technique for solving. NSSR manifested promising recovery performance compared to the other existing approaches. However, most of these methods aimed at recovering the HR-HS image via minimizing the reconstruction errors of the observed HR-RGB and LR-HS images in couple, thus requiring the precise alignment of the HR-RGB and LR-HS images, where most available images cannot satisfy the strict situation. In [13], Akhtar et al. explored a sparse spatio-spectral representation via assuming the same atoms used for reconstructing the spectra of the pixels in a local grid region and proposed a generalized simultaneous orthogonal matching pursuit (G-SOMP+) method for estimating the sparse coefficients. The G-SOMP+ method can integrate the spectral correlation property in the LR-HS image and the HR spatial structure in the HR-RGB image, but does not need the precise alignment between the observed images. Although simultaneously selecting the common set of spectral bases for the pixel spectra in a grid of a local region can take the spectral similarity into consideration, the size of the selected base set is the same for all pixels in the local region and not always the optimized one for approximating all of the pixel spectra. Our study is most related to the work [13]. However, we select the support set of the previously learned spectral dictionary for each pixel only one time, where the size of the support set is adaptively adjusted with the spectral mixing degree according to local content similarity analysis in the observed HR-RGB image.

## 3. Problem Formulation

Our goal is to estimate an HR-HS image $Z^{\prime }\in R^{W\times H\times L}$, where *W* and *H* denote the spatial dimensions and *L* is the spectral channel number, from an LR-HS image $X^{\prime }\in R^{w\times h\times L}$ (w≪W, h≪H) and an HR-MS (RGB) image $Y^{\prime }\in R^{W\times H\times l}$ (l≪L). Since the observed HR-MS image is an RGB image, the spectral channel number *l* in the HR-RGB image is three. We reformulate $Z^{\prime }$, $X^{\prime }$, and $Y^{\prime }$ as the pixel-wise spectral representations (matrix forms), denoted as $Z\in R^{L\times N}$ (N=W×H), $X\in R^{L\times O}$ (O=w×h), and $Y\in R^{3\times N}$, respectively. Both the matrix forms X and Y of the LR-HS and HR-RGB images can be expressed as a linear combination of the HR-HS image Z:(2)X=ZD,Y=RZ,
where $D\in R^{N\times O}$ is the blurring and down-sampling operators on the HR-HS image to generate the LR-HS image X, and $R\in R^{3\times L}$ represents the spectral response function (transformation matrix) decided by the camera design, which maps the HR-HS image Z to the HR-RGB image Y. Since the number of the unknowns (NL) in the desired HR-HS image is much larger than the total number of measurements (OL+3N) from the observed images X and Y, the HS image superresolution is a severely ill posed problem. In order to provide a stable estimation of the HR-HS image, regularized constraints for integrating the prior knowledge about the unknown Z are widely explored to narrow the solution space. A general strategy is to assume that the pixel spectrum in the HR-HS image lies in a low-dimensional space and can be decomposed into the spectral reflectance of several pure materials and their corresponding composition fractions as follows:(3)$$\begin{array}{cc} z_{n} & =\sum_{k=1}^{K}b_{k}\alpha_{k,n}=B\alpha_{n}\\ \; & \;\mathrm{subject}\;\mathrm{to}:\;\;b_{i,k}\geq 0,\alpha_{k,n}\geq 0,\sum_{k=1}^{K}\alpha_{k,n}=1, \end{array}$$
where $B\in R^{L\times K}$ represents the spectral response functions (spectral signatures) of *K* distinct materials (also called endmembers) in the observed LR-HS image and $\alpha_{n}$ denotes the fractional abundance of the *K* materials in the covered area for the *n*th pixel. Taking the physical phenomenon of the spectral reflectance into consideration, the elements in the spectral endmember and the fraction magnitude of the abundance are non-negative, as shown in the first and second constraint terms. The abundance vector for each pixel is summed to one, where each element means the composite percentage of a distinct material. The above formulation of the spectral representation is popularly known as the linear mixture model [33]. Since the regularized term $\sum_{k=1}^{K}\alpha_{k,n}=1$ with a nonnegative constraint is equivalent to the $l_{1}$ norm, which is a relaxation of the $l_{0}$ norm for the sparse constraint, Equation (Equation 3) can also be considered as a sparse spectral representation via alleviating the limitation of the spectral atom number. The observed HR-RGB image can be formulated as Y=RZ, and each pixel $y_{n}\in R^{3}$ is written as:(4)$$y_{n}=Rz_{n}=RB\alpha_{n}=B^{l}\alpha_{n},$$
where $B^{l}$ is the RGB spectral dictionary, which is obtained via transforming the HS dictionary B with the spectral transformation matrix R: $B^{l}=RB$. With a corresponding set of the spectral dictionaries $B^{l}$ and B, the sparse fractional vector $\alpha_{n}$ of the HS pixel $z_{n}$ can be predicted from the HR-RGB pixel $y_{n}$. We reformulate Equation (Equation 3) as the objective function for minimization via replacing the known RGB vector in the HR-RGB image:(5)$$\underset{A}{\arg\;\min}{∥A∥}_{1},\;\mathrm{subject}\;\mathrm{to}:\;{∥Y-B^{l}A∥}_{F}\leq \eta ,A\geq 0$$

### G-SOMP+ Method

Akhtar et al. [13] proposed a generalization of a popular greedy pursuit algorithm, called generalized simultaneous orthogonal matching pursuit (G-SOMP+) for optimizing Equation (Equation 5). Motivated by the fact that nearby pixels are likely to represent the same materials in the scene, the G-SOMP+ method processes the HR-RGB image Y in terms of small disjoint spatial patches for computing the coefficient matrix. Denoted the HR-RGB image patch by $P^{\prime }\in R^{W_{O}\times H_{P}\times 3}$ with matrix representation $P\in R^{3\times W_{O}H_{P}}$, G-SOMP+ estimates its corresponding coefficient matrix $A_{P}\in R^{l\times W_{O}\times H_{P}}$ via solving the constrained simultaneous sparse approximation problem:(6)$$\underset{A_{P}}{\arg\;\min}∥A_{P}{∥}_{row\_0},\;\mathrm{subject}\;\mathrm{to}:\;{∥P-B^{l}A_{P}∥}_{F}\leq \eta ,\alpha_{P^{i}}\geq 0$$
where $\alpha_{P^{i}}$ denotes the *i*th column of the matrix $|A_{P}$. In Equation (Equation 6), $∥A_{P}{∥}_{row\_0}$ is the row $l_{0}$ quasi-norm [34] of the matrix, which represents the cardinality of its row support, formulated as:(7)$$∥A_{P}{∥}_{row\_0}\Leftrightarrow \left|\cup_{i=1}^{W_{P}H_{P}}Supp\left(\alpha_{P^{i}}\right)\right|$$
where supp(·) denotes the support of a vector and |·| indicates the cardinality of a set. G-SOMP+ was proposed for optimizing the objective function in Equation (Equation 6) and allows the selection of multiple dictionary atoms in each iteration of orthogonal matching pursuit for efficient computation. The algorithm pursues an optimized approximation of the input spectral matrix P, the spectral representation of a local patch, by selecting the dictionary atoms $B_{P}^{l}$ indexed in a set Ξ∈Ω (Ω is the index space of the RGB dictionary $B^{l}$), such that Ξ≪Ω, and the selected $B_{P}^{l}$ contributes to the approximation of all the pixel spectra in the local patch. Similar to orthogonal matching pursuit (OMP), G-SOMP+ also exploits the iteration procedure for calculating the sparse vector. In the *i*th iteration, firstly, the algorithm computes the cumulative correlation of each dictionary atom with the residue of its current approximation of all pixel spectra in the focused local patch, where the initial residue in the first iteration is the pixel spectra themselves in the patch. Then, *M* dictionary atoms with the highest cumulative correlations are selected and are added to an index subspace: Ξ, which is empty at initialization, of the index set: Ω. Finally, the dictionary atoms in the aforementioned index subspace: Ξ are used for a non-negative least squares approximation of the patch. At the same time, the residue is updated for the next iteration and judging if the algorithm would stop or not according to a fraction γ of the residue in the previous iteration. Although the G-SOMP+ method is able to select different numbers of dictionary atoms for different patches, the section criteria pursue the common dictionary atoms for all pixel spectra in a local patch, which is not always optimized for all pixels. Furthermore, the highest cumulative correlations to the residual not the raw pixel spectra cannot guarantee the good reconstruction of the raw pixels.

## 4. Data Guided Sparsity Regularized Spectral Representation

The matrix representation of the spectral representation model in Equations (3) and (4) can be rewritten as:(8)$$Z=BA,\;Y=B^{l}A,$$
where $A=\left[\alpha_{1},\alpha_{2},\cdots ,\alpha_{K}\right]\in R_{+}^{K\times N}$ is the non-negative sparse coefficient matrix. Both spectral dictionary B or $B^{l}$ and coefficient matrix A are unknown, and regularization for exploring the prior knowledge of the unknown is often integrated to give a stable solution. As in the work [13], we firstly learn the HS dictionary from the observed LR-HS image via a hierarchical clustering method with correlation similarity as elaborated in the next subsection such that the spectral sample can be represented by a weighted combination of the learned over-complete spectral dictionary $B\in R^{L\times K}\left(K>L\right)$. Then, the transformed RGB dictionary $B^{l}=RB$ can be used for estimating the sparse vector using Equation (Equation 5) for each HR RGB pixel, which in turn can approximate the HR hyperspectral pixel with the HS dictionary B. However, the conventional sparse methods usually explore an identical strength of constraints on all the spectral samples without considering the individual property, which may not meet the practical situation. This study proposes a data guided sparsity regularized nonnegative spectral representation for the hyperspectral superresolution task. Motivated by the fact that the material purity of each pixel spectrum might be different from the others, we firstly explore the possible purity (mixing degree) of each pixel according to the local content similarity of a focused pixel and incorporate the material purity as the sparsity constraint into the non-negative sparse spectral representation, which adaptively imposes the sparse constraint for each pixel. Next, we will introduce how to learn the HS dictionary B, calculate the data guided sparsity for each pixel, and optimize the sparse coefficient matrix A in our method.

### 4.1. On-Line Hyperspectral Dictionary Learning

Due to the large variety of HS reflectance for different materials, learning a common HS dictionary for different observed scenes with different materials leads to considerable spectral distortion for the approximated pixel spectra. This study proposes to generate the HS dictionary from the observed LR-HS image of the under-studied scene. It is known that the raw hyperspectra in the observed LR-HS image can be directly used as the HS dictionary and then are transformed into the RGB bases for calculating the sparse representation of all the pixels in the observed HR-RGB image. However, in spite of the low resolution in the observed HS image, it is still possible to have some similar spectral pixels, and this leads to redundancy in the HS dictionary. Therefore, in order to reduce the redundancy of the HS bases via directly using the raw hyperspectra, we propose an online dictionary learning algorithm for generating more compact hyperspectral bases. Since different scenes may be composed of different numbers of materials, we do not fix the number of the learned spectral bases in the dictionary and adaptively decide the bases’ number according to the similarity degree of the spectra in the observed scene. The detail procedure of the on-line dictionary learning is manifested in Algorithm 1. Algorithm 1 is implemented via clustering high correlated spectra with a pre-defined threshold 0.999 and then adaptively generating a different number of HS bases according to the contents of the available LR-HS images.

After the HS dictionary B is learned online from the observed HR-HS image, the corresponding multispectral dictionary $B^{l}$ is then obtained by transforming B with the camera spectral response matrix R as $B^{l}=RB$. Via fixing the learned HS and RGB dictionaries B, $B^{l}$, we aim to calculate the sparse coefficient matrix A with the available HR-RGB image Y with the data guided sparsity constraint.

**Algorithm** **1** On-line hyperspectral dictionary learning algorithm.Initialization: Taking the HS reflectance of all pixels in the observed LR-HS image as the HS sample set {S}, initialize dictionary set B as {ϕ}, and set the correlation similarity threshold θ;1. Randomly select an HS reflectance $s_{t}$ as temporary set $\left\{S^{t}\right\}=\left\{s_{t}\right\}$ from {S}, and update {S} by removing $s_{t}$;2. Calculate the normalized correlation coefficients $r_{i}$ between $s_{t}$ and any sample $s_{i}$ in {S}:               $r_{i}=\frac{\sum_{l=1}^{L}s_{t,l}s_{i,l}}{\sqrt{\sum_{l=1}^{L}s_{t,l}}\sqrt{\sum_{l=1}^{L}s_{i,l}}}$If $r_{i}>\theta$, we take $s_{i}$ from {S} into $\left\{S^{t}\right\}$
$\left\{S\right\}\Rightarrow^{s_{i}}\left\{S^{t}\right\}$.3. Repeat the same process for all samples in {S}4. Calculate the mean vector in $\left\{S^{t}\right\}$ as the HS basis $b_{t}$; put it into the dictionary B
$b_{t}\Rightarrow B$; and re-set $\left\{S^{t}\right\}$ as ϕ.6. If {S} is ϕ, finish; otherwise go to Step 1.

### 4.2. Calculation of the Data Guided Sparsity Map

The data guided sparsity map is trained from the HR-RGB image that describes the strength of priors (constraints) for each individual sample via measuring the mixed level of each pixel. The calculation of the sparsity for each pixel is motivated by the observation that the mixed level of each pixel should be different from each other with a high mixed level in the transition area and a low mixed level in the smoothed area. It can be calculated by the similarity of the spatially neighboring pixels, and the greater the mixed level, the weaker sparsity of the pixels in the transition area.

Given the available HR-RGB image Y with *N* pixels and *L* channels, we measure the uniformity of neighboring pixels as the adaptive sparsity prior, which is the inverse of the mixed level, over the entire image. For the *i*th pixel, its sparsity measurement p(i) could be estimated by exploring the similarity between spatially neighboring pixels as follows:(9)$$p\left(i\right)=\sum_{j\in NN_{i}}s_{ij}$$
where $NN_{i}$ is the neighborhood of the *i*th pixel that includes four neighbors; $s_{ij}$ is the similarity between the *i*th pixel and its neighboring pixel $y_{j}$ by the dot-product metric:(10)$$s_{ij}=\exp\left(-\frac{∥y_{i}-y_{j}{∥}^{2}}{\sigma }\right)$$

The value of σ controls the smoothness of the sparsity map. Some examples of the calculated sparsity maps for three images are given in Figure 1, which shows that the smoothed regions in the input RGB images have large magnitudes (strong sparsity) in the sparsity maps and the transition areas have small magnitudes (weak sparsity).

### 4.3. The Non-Negative Sparse Vectors with the Adaptive Sparsity

This section estimates the non-negative sparse vector A by optimizing the objective function: $\min_{A}{\left\|Y-B^{l}A\right\|}_{F}^{2}$ with sparsity constraint and non-negativity on A. The generally used constraint for sparsity is the $l_{0}$ norm: $\|\alpha_{n}^{t}{\|}_{0}<\eta$, which is implemented by matching pursuit methods and needs the computational cost as $O(K^{2})$ (Kis the atom number in the dictionary). On the other hand, Yu et al. [35,36] empirically observed that SCresults tend to be local, which means nonzero coefficients are often assigned to bases near the encoded data, and proposed a locality constrained linear coding method for explicitly encouraging local encoding and efficiently approximation with computational cost: O(K). This study explores a nonnegative locality constrained linear coding for obtaining the sparse vector of an RGB spectral $y_{n}$ (the RGB spectral of the *n*th pixel in Y) in the observed HR-RGB image as follows:(11)$$\begin{array}{c} \min_{\alpha_{n}}{\left\|y_{n}-B_{NN(y_{n})}^{l}\alpha_{n}\right\|}^{2}\\ \mathrm{subject}\;\mathrm{to}:\;B_{NN(y_{n})}^{l}\;\mathrm{are}\;\mathrm{the}\;K^{\prime }\;\mathrm{nearest}\;\mathrm{bases}\;\mathrm{of}\;y_{n},\\ \alpha_{k,n}>=0, \end{array}$$

In the above equation, we firstly use the Euclidean distance for calculating the *K* nearest bases from B for the input sample $y_{n}$ and apply the non-negative least squares algorithm to calculate the non-negative coefficient for the selected basis only, which are just a small part of the whole basis B. Generally, the selected basis number $K^{\prime }$ is previously defined and is fixed for all samples. As analyzed in the above section, since the sparsity for different samples should be different, we adaptively set the selected basis numbers for different pixels according to the calculated sparsity map. Given the processed images, we learned *K* HS bases for representing spectral pixels and calculated the sparsity p(i) for the *i*th pixel. The selected basis number for the *i*th pixel can be adaptively set as:(12)$$K^{\prime }=M*\exp\left\{-\left[p\left(i\right)-\frac{1}{N}\sum_{j=1}^{N}p\left(i\right)\right]\right\}$$
where *N* is the pixel number in the processed image and *M* is a hyper-parameter, which is pre-defined and can be considered as the average number of dictionary atoms used for all pixels in the under-studied scene. With Equation (Equation 12), we can adaptively calculate the required basis number $K^{\prime }$ for representing the spectrum of each pixel.

## 5. Experimental Results

We evaluated the proposed approach using two publicly available HS imaging database: the CAVEdataset [37] and the Harvard dataset [38], and three real satellite images including the Hyperspec-VNIR Chikusei image [39], Salinas, and University of Pavia scenes [40,41,42]. The CAVE dataset consists of 32 indoor images including paintings, toys, food, and so on, captured under controlled illumination, and the Harvard dataset has 50 indoor and outdoor images recorded under daylight illumination. The dimensions of the images from the CAVE dataset are 512×512 pixels, with 31 spectral bands of 10 nm wide, covering the visible spectrum from 400 to 700 nm; the images from the Harvard dataset have the dimensions of 1392×1040 pixels with 31 spectral bands of a width of 10 nm, ranging from 420 to 720 nm, from which we extracted the top left 1024×1024 pixels in our experiments. The Hyperspec-VNIR Chikusei image is an airborne HS dataset, which was taken by Headwall’s Hyperspec-VNIR-C imaging sensor over Chikusei, Ibaraki, Japan, on 29 July 2014. The dataset comprises 128 bands in the spectral range from 0.363 to 1.018 μm. The scene consists of 2517×2335 pixels with a GSD of 2.5 m, mainly including agricultural and urban areas. A selected 540×420 pixel size image was used in the experiment. The Salinas image comprising 512×217 pixels with 220 bands was also collected by the AVIRIS sensor, capturing an area over Salinas Valley, California, with a spatial resolution of 3.7 m. The University of Pavia scenes (PaviaU) were collected by the Reflective Optics System Imaging Spectrometer (ROSIS) sensor. This image with 610×340 pixels and 103 spectral bands covering the city of Pavia, Italy, was collected under the HySens project managed by DLR (the German Aerospace Agency).

For the CAVE and Harvard datasets, we treated the original images in the databases as the ground truth Z and down-sampled them by a factor of 32 to create 16×16 images, which was implemented by averaging over 32×32 pixel blocks, as done in [13,27]. The observed HR-RGB images Y were simulated by integrating the ground truth over the spectral channels using the spectral response R of a Nikon D700 camera. For the Hyperspec-VNIR Chikusei data, we used the released LR-HS (size: 90×70×128) and HR-MS (size: 540×420×8) images by Yokoya [39] as the input and estimated an HR-HS image with a size of 540×420×128. Since the the camera spectral response function R for the generation the MS image from the HS image is unknown, we exploited the quadratic programming method for automatically estimating R. Since Salinas and PaviaU data are generally used for HS image classification, there are no corresponding HR multi-spectral (MS) images and LR-HS images for our HS image superresolution scenario. We considered the Salinas and PaviaU data as the ground truth HR-HS images and simulated the corresponding LR-HS images via simply average down-sampling with a factor of: six for both the horizontal and vertical directions and the eight-band HR-MS images via generating the spectral response with Gaussian functions. Then, we adopted different HS image superresolution methods on the simulated LR-HS and HR-MS images to estimate the HR-HS image for evaluation.

To evaluate the quantitative accuracy of the estimated HS images, four objective error metrics including root-mean-squared error (RMSE), peak-signal-to-noise ratio (PSNR), relative dimensionless global error in synthesis (ERGAS) [43], and spectral angle mapper (SAM) [9] were evaluated. The metrics: ERGAS [43] calculates the average amount of the relative square error, which is normalized by the intensity mean in each band as defined below:(13)$$\begin{array}{c} \mathrm{ERGAS}=100\times \frac{O}{N}\sqrt{\frac{1}{L}\sum_{l=1}^{L}\left(\frac{\mathrm{MSE}(i)}{\mu_{i}}\right)} \end{array}$$
where $\frac{O}{N}$ is the ratio between the pixel sizes of the available HR-RGB and LR-HS images, $\mu_{i}$ is the intensity mean of the *i*th band of the LR-HS image, and *L* is the band number in the LR-HS image. The smaller the ERGAS, the smaller the relative error in the estimated HR-HS image. SAM [9] measures the spectral distortion between the ground truth and the estimated HR-HS images, and the distortion of two spectral vectors $z_{n}$ and ${\hat{z}}_{n}$ is defined as follows:(14)$$\begin{array}{c} \mathrm{SAM}\left(z_{n},{\hat{z}}_{n}\right)=\arccos\left(\frac{<x_{n},{\hat{z}}_{n}>}{\|z_{n}{\|}_{2}{\left\|{\hat{z}}_{n}\right\|}_{2}}\right) \end{array}$$

The overall SAM was finally obtained by averaging the SAMs computed from all image pixels. Note that the value of SAM is expressed in degrees and thus belongs to (−90, 90]. The smaller the absolute value of SAM, the less important the spectral distortion.

As introduced in Section 4.1, we trained the HS dictionary from the observed LR-HS image for the under-studied scene instead of learning the common dictionary for different scenes. The histogram of the dictionary number of all images in the CAVE and Harvard databases is shown in Figure 2a,b, respectively, which manifested a much smaller number of learned bases than the raw spectral numbers (256 for the images in CAVE and 1024 in Harvard). As we know that the computational cost of the sparse coding procedure in Equation (Equation 5) is proportional to the basis number used, thus the calculating time of the sparse coding would be greatly reduced for the images in both the CAVE and Harvard datasets. The online dictionary learning (denoted as OLD) took much less than 0.01 s, which is much less than the computational time of the sparse coding. Since our proposed method explored the spectral representation with the pixel-wise sparsity constraint, we modified the G-SOMP+ [13] method for selecting the dictionary atom of each pixel separately instead of the whole patch for fair comparison, called pixel-wise G-SOMP+, where the hyper-parameter *M* was the predefined number of the simultaneously selected dictionary atoms. We also fixed the number of used dictionary atoms for the pixel spectral representation and adaptively adjusted the used atom number according to our proposed data guided sparsity map. We conducted experiments on all images in the CAVE and Harvard datasets and calculated the average RMSE, PSNR, SAM, and ERGAS for comparison. The compared quantitative results with different hyper-parameters *M* are shown in Figure 3 and Figure 4 for the CAVE and Harvard datasets, respectively. From Figure 3, it can be seen that the pixel-wise G-SOMP+ could achieve better performance than the fixed number of the used dictionary atoms for spectral representation (denoted “Fixed”), and our proposed method with data guided sparsity could further improve the performance for most hyper-parameters and all quantitative metrics in the CAVE dataset. For the Harvard dataset, the “Fixed” number of the used dictionary atoms for spectral representation provided better results than the G-SOMP+ method, and our proposed method could further improve the performance, especially for the ERGAS metric. In addition, we provide the computational time of the pixel-wise G-SOMP+, the spectral representation with the fixed number, and the data guided number of the dictionary atoms in Figure 5. Figure 5 manifests that our method had comparable computational cost with the fixed number of dictionary atoms, but about seven times faster than the pixel-wise G-SOMP+ method.

Next, we show the performance of our proposed method, compared with the state-of-the-art HSI SR methods including: the matrix factorization method (MF) method [27], coupled non-negative matrix factorization (CNMF) method [28], sparse non-negative matrix factorization (SNNMF) method [44], generalization of simultaneous orthogonal matching pursuit (G-SOMP+) method [13], and Bayesian sparse representation (BSR) method [9]. The average RMSE, PSNR, SAM, and ERGAS results of the 32 recovered HR-HS images from the CAVE dataset [37] are shown in Table 1, while the average results of the 50 images from the Harvard dataset [38] are given in Table 2.

From Table 1 and Table 2, we observe that for all error metrics, our approach achieved better than or comparable performance as the state-of-the-art methods. It should be noted that most existing methods excepting the G-SOMP+ approach [13] simultaneously (coupled) minimized the reconstruction errors of the available HR-RGB and LR-HS images. Since we attempted to validate the efficiency of the adaptive sparsity strategy for different samples, our proposed method simply optimized the sparse representation with the available HR-RGB image only like in the G-SOMP+ approach [13] and can be extended to any other coupled method for more accurate performance. Figure 6 shows an example of the recovered HR-HS images (the image of the 20th band) with the pixel-wise G-SOMP+ method, the fixed, and the adaptive basis number and the corresponding difference image between the ground truth and the recovered images in the CAVE dataset, while Figure 7 manifests an example of the recovered HR-HS image (the image of the 20th band) with the pixel-wise G-SOMP+ method, our proposed method, and the corresponding difference image between the ground truth and the recovered images in the Harvard dataset.

Finally, we provide the compared results with the pixel-wise G-SOMP+ method, the spectral representation with the fixed number, and the data guided number of the dictionary atoms on three real satellite images including the Hyperspec-VNIR Chikusei image, PaviaU image, and Salinas image. The compared quantitative results are manifested in Table 3. Table 3 also validates that our proposed method with the data guided sparsity could not only improve the performance, but also reduced the computational time compared with the pixel-wise G-SOMP+ method. The recovered HS images (the pseudo-color images with the 72nd, 76th, and 80th bands) and the absolute difference images with respect to the ground truth in an expanded region are visualized in Figure 8, which also manifests the smaller difference error in our proposed method with the data guided sparsity. The input HS images and the recovered HS images for Salinas and PaviaU data (the pseudo-color images with the 72nd, 76th, and 80th bands for Salinas data and the 22nd, 100th and 170th for PaviaU data) are given in Figure 9.

## 6. Conclusions

This study proposed a novel data guided sparse spectral representation method for the HS image superresolution task. The proposed method firstly learned the HS dictionary using the available LR-HS image and then transformed it into the RGB one with the camera response function. The non-negative sparse representation of each pixel in the HS image could be calculated given the RGB vector and the RGB dictionary. The state-of-the-art methods generally calculate the sparse vector with the previously defined sparsity strength regardless of the property of different samples. This study investigated the spectral property via analyzing the local content similarity of a focused pixel in the available high resolution RGB image and generated a sparsity map for guiding the calculation of the sparse spectral representation. Motivated by the fact that the pixel spectrum of a pure material should have more sparse representation of the spectral dictionary, we quantified the spectral mixing degree via measuring the similarity of a pixel with the surrounding pixels and then controlled the sparsity in the computation of the spectral representation. Since the proposed method adaptively adjusted the sparsity in the spectral representation based on the local content of the available high resolution RGB image, it could produce more robust spectral representation for recovering the target high resolution hyperspectral image. Comprehensive experiments on two public HS datasets and three real remote sensing images validated that the proposed method achieved promising performances compared with the existing state-of-the-art methods.

## Figures and Tables

**Figure 1 sensors-19-05401-f001:**
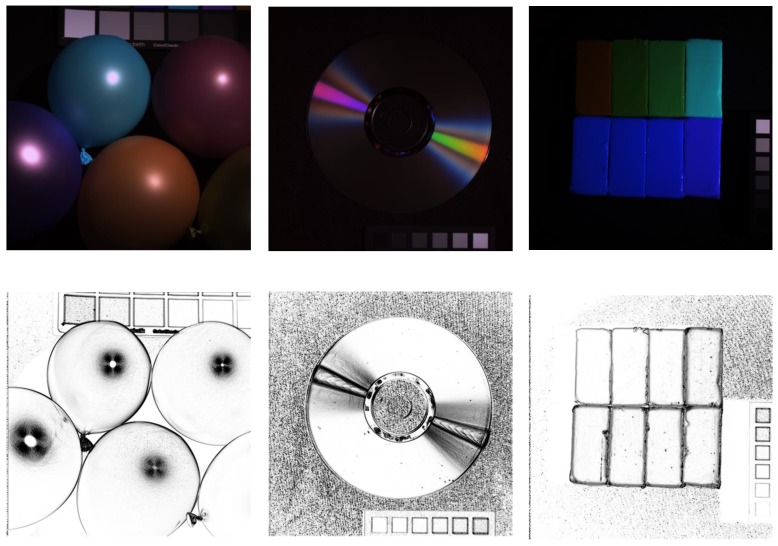
Three examples of the calculated sparsity maps. The first row manifests the RGB images, and the second row gives the calculated sparsity maps.

**Figure 2 sensors-19-05401-f002:**
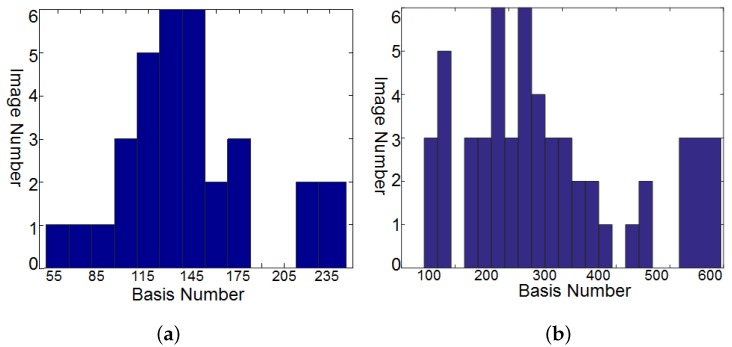
The histogram of the dictionary number of all images in the CAVEand Harvard databases. (**a**) Cavedataset; (**b**) Harvard dataset.

**Figure 3 sensors-19-05401-f003:**
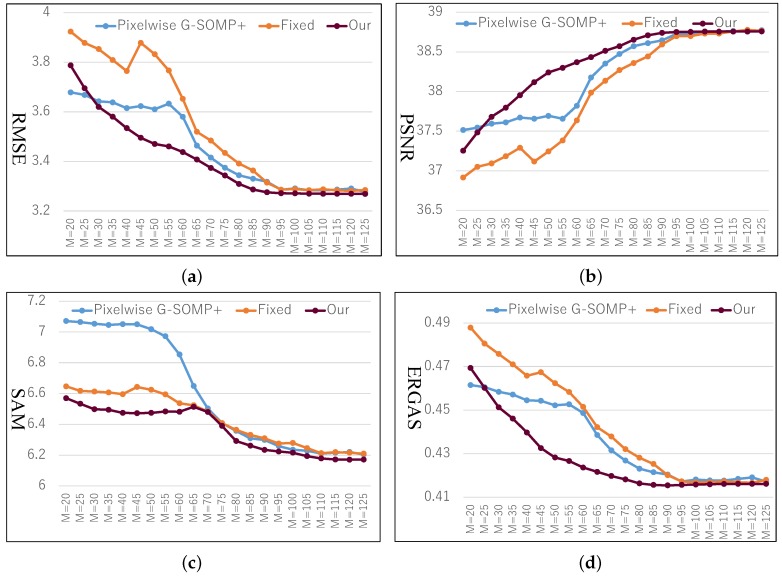
Quantitative compared results with the pixel-wise generalized simultaneous orthogonal matching pursuit (G-SOMP+) method, the spectral representation of the fixed number, and the data guided number of the dictionary atoms (our proposed method) under different hyper-parameters *M* on the CAVE dataset. (**a**) RMSE; **(b**) PSNR; (**c**) spectral angle mapper (SAM); (**d**) relative dimensionless global error in synthesis (ERGAS).

**Figure 4 sensors-19-05401-f004:**
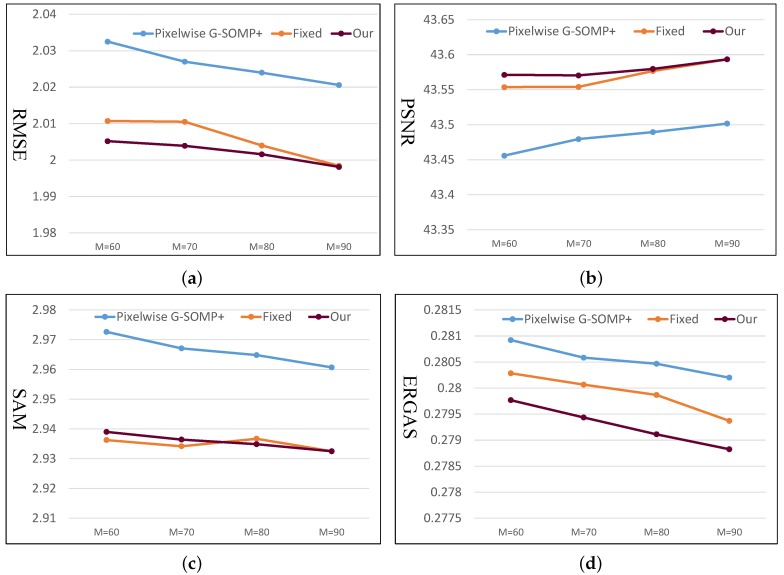
Quantitative compared results with the pixel-wise G-SOMP+ method, the spectral representation of the fixed number, and the data guided number of the dictionary atoms (our proposed method) under different hyper-parameters *M* on the Harvard dataset. (**a**) RMSE; (**b**) PSNR; (**c**) SAM; (**d**) ERGAS.

**Figure 5 sensors-19-05401-f005:**
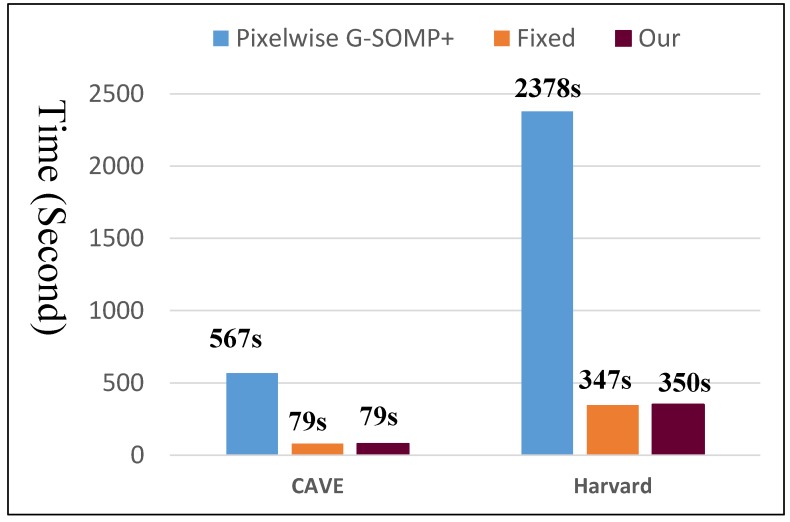
The compared computational time for both the CAVE and Harvard datasets.

**Figure 6 sensors-19-05401-f006:**
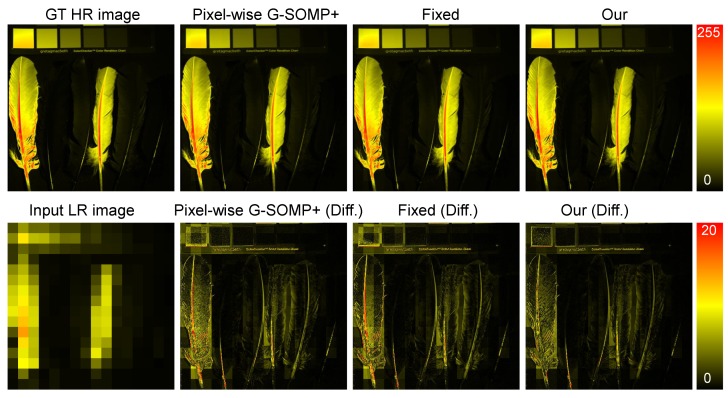
An example of the recovered HR-HS images with the pixel-wise G-SOMP+ method, the fixed and the adaptive basis number, and the corresponding difference images between the ground truth and the recovered images from the CAVE dataset.

**Figure 7 sensors-19-05401-f007:**
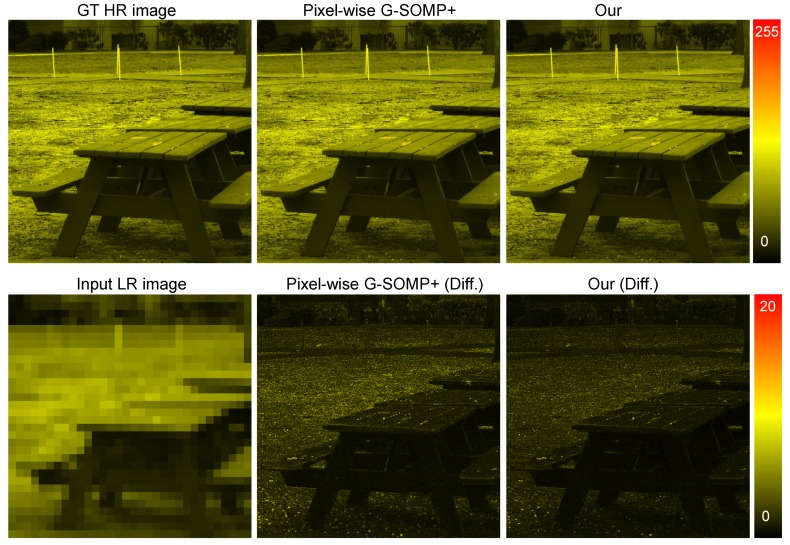
An example of the recovered HR-HS images with the pixel-wise G-SOMP+ method and the corresponding difference images between the ground truth and the recovered images from the Harvard dataset.

**Figure 8 sensors-19-05401-f008:**
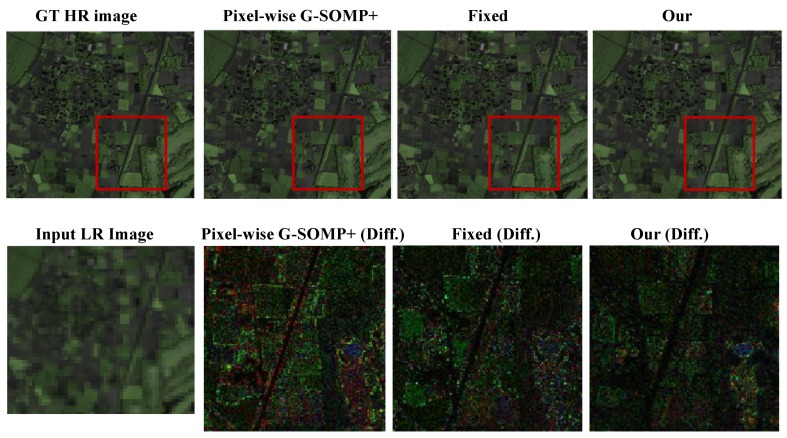
The recovered HR-HS images on the Hyperspec-VNIR Chikusei data with the pixel-wise G-SOMP+ method, the fixed and adaptive basis number, and the absolute difference images in an expanded regions between the ground truth and the recovered images.

**Figure 9 sensors-19-05401-f009:**
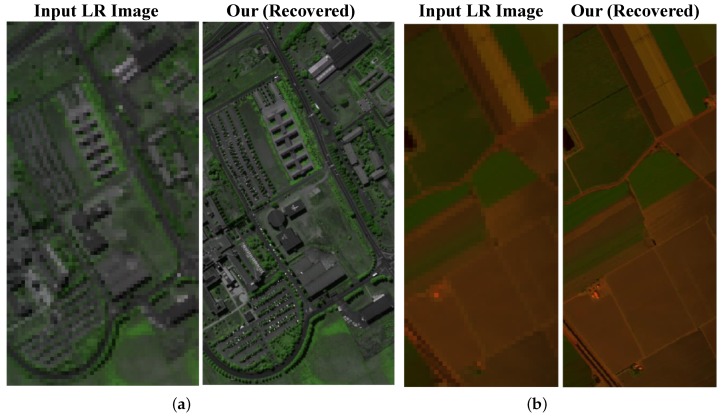
The input HR-HS image and the recovered HR-HS images on the Salinase and PaviaU images. (**a**) Salinas image. (**b**) PaviaU image.

**Table 1 sensors-19-05401-t001:** Quantitative compared results on the CAVE dataset. matrix factorization method (MF) method, coupled non-negative matrix factorization (CNMF) method, sparse non-negative matrix factorization (SNNMF) method, generalization of simultaneous orthogonal matching pursuit (G-SOMP+) method, and Bayesian sparse representation (BSR) method.

	RMSE	PSNR	SAM	ERGAS
MF [27]	3.03 ± 1.44	39.37 ± 3.76	6.12 ± 2.17	0.40 ± 0.22
CNMF [28]	2.93 ± 1.30	39.53 ± 3.55	5.48 ± 1.62	0.39 ± 0.21
SNNMF [44]	3.26 ± 1.57	38.73 ± 3.79	6.50 ± 2.32	0.44 ± 0.23
GSOMP [13]	6.47 ± 2.53	32.48 ± 3.08	14.19 ± 5.42	0.77 ± 0.32
BSR [9]	3.13 ± 1.57	39.16 ± 3.91	6.75 ± 2.37	0.37 ± 0.22
Ours	3.27±1.63	38.76 ± 4.00	6.17 ± 2.17	0.41 ± 0.22

**Table 2 sensors-19-05401-t002:** Quantitative compared results on the Harvard dataset.

	RMSE	PSNR	SAM	ERGAS
MF [27]	1.96 ± 0.97	43.19 ± 3.87	2.93 ± 1.06	0.23 ± 0.14
CNMF [28]	2.08 ± 1.34	43.00 ± 4.44	2.91 ± 1.18	0.23 ± 0.11
SNNMF [44]	2.20 ± 0.94	42.03 ± 3.61	3.17 ± 1.07	0.26 ± 0.27
GSOMP [13]	4.08 ± 3.55	38.02 ± 5.71	4.79 ± 2.99	0.41 ± 0.24
BSR [9]	2.10 ± 1.60	43.11 ± 4.59	2.93 ± 1.33	0.24 ± 0.15
Ours	1.99±1.35	43.59 ± 4.85	2.96 ± 1.08	0.27 ± 0.15

**Table 3 sensors-19-05401-t003:** Quantitative compared results using three real satellite images.

(a) The Hyperspec-VNIR Chikusei Image					
	RMSE	PSNR	SAM	ERGAS	Time (s)
Pixel-wise G-SOMP+	2.68	39.57	3.42	2.85	268.84
Fixed	3.35	37.63	4.35	2.96	94.21
Ours	2.50	40.17	3.19	2.80	95.94
**(b) The Salinas Image**					
	RMSE	PSNR	SAM	ERGAS	Time (s)
Pixel-wise G-SOMP+	1.97	42.24	2.18	0.97	267.57
Fixed	2.15	41.48	2.39	1.06	81.87
Ours	1.96	42.30	2.18	0.97	83.76
**(c) The PaviaU Image**					
	RMSE	PSNR	SAM	ERGAS	Time (s)
Pixel-wise G-SOMP+	1.04	47.75	1.18	1.29	155.47
Fixed	1.07	47.35	1.20	1.17	47.57
Ours	1.03	47.80	1.17	1.09	48.67

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
