# Peer review of "Spectral Representation via Data-Guided Sparsity for Hyperspectral Image Super-Resolution†"

_sensors, 2019, doi:10.3390/s19245401_

Round 1

Reviewer 1 Report

This paper proposes a hyperspectral image (HSI) super-resolution method using observed HSI with low spatial resolution and RGB image with low spectral resolution. A sparsity map learned from the observed HSI is proposed and is used to guid the sparsity of the estimates of dictionary coefficients. The motivation is explained clearly. But there are some issues that need to be addressed before its publication.

The authors claim that given the HSI dictionary and RGB dictionary, the representation coefficients of a hyperspectral vector can be predicted from the RGB vector. Is it always true? What happens if a pixel is composed of more than 3 materials? The proposed method may not be suitable for highly mixed HSIs.

There are many typos and mistakes. 

Page 3, Line 113: Y’

Page 4, Line 115: Y’<— Y

Page 4, Line 129: R=RB

Page 4, Line 139 and the title: vis<— via

Page 6, Line 181: spares <— sparse

Page 8, eq. (11) : min_{ffn}

Algorithm 1: r_i should be greater than /theta if it is looking for similar spectral vectors. The formulation of normalized correlation coefficients is wrong in Algorithm 1.

Author Response

Reply to Expert Reviewer 1

1. The authors claim that given the HSI dictionary and RGB dictionary, the representation coefficients of a hyperspectral vector can be predicted from the RGB vector. Is it always true? What happens if a pixel is composed of more than 3 materials? The proposed method may not be suitable for highly mixed HSIs. (The first comment of Expert reviewer 1)

Response: Thanks for pointing out this issue. We are respectfully agreed with the reviewer that the representative coefficients of a hyperspectral vector are not always same to the coefficients of the corresponding RGB vector. However, the hyperspectral vector in HS superresolution scenario is not available for computing its coefficients, and thus we assume the representation coefficients of a HS vector can be predicted from the RGB vector given the correspondence between the HS and RGB dictionary, which are similar to the assumed consistent sparse vector between LR and HR patches in example-based image superresolution such as SRSC (Jianchao Yang etc., Super-Resolution via Sparse Representation). Of course, we can also learn a mapping function between the coefficients of the HR and RGB vectors. However, the mapping function learning needs additional training RGB and HS vector samples, and the previously learned mapping function may not always match the relation of the test samples. Therefore, this study simply assumes the similar representation coefficients of the HS and RGB vector given the corresponding HS and RGB dictionary. We evaluate the our method on the downsampled LR-HS image (Which would be highly mixed HSIs: CAVE and Harvard datasets) with large factor such as 32 in horizontal and vertical directions, respectively, and have proven promising performance (Experimental results section). Furthermore, we allow the nonzero elements in the representation vector more than 3, and thus the pixel spectrum composed of more than 3 materials can be still accurately approximated in our method.

2. There are many typos and mistakes.  (Comments from the Expert reviewer 1)

Response: Thanks for helping us refine our paper. I modified typos and mistakes according reviewer’s comments. Thanks.

Reviewer 2 Report

This work similar to their previous work tries to explore the idea to generate HR images from HS or MS data. The work is quite interesting however there few concerns that I believe deserve to be addressed.

To some extent, I feel the authors failed to justify the difference between their previous and current work. Extensions are not clearly explained! The problem statement is not clear and not well defined which lacks the novelty content. The Intuition regarding the pure spectral is obvious, lines 51-52. What's new in this?

There are many equations need to be revised, e.g. eq. 14. 

Extensive Englished editing required. There are many long sentences that are really hard to grasp, e.g. Line 33-38, 44-47. Many acronyms defined but never used e.g. Hyperspectral (HS) is defined however the full form is being used in several places. 

Experimental results are not very convincing, as there are many HSI datasets existed which should also be tested. Such as the well known HSI datasets for Unmixing and classification used in several state-of-the-art works such as, "Unsupervised geometrical feature learning from hyperspectral data, IEEE SSCI", "Hyperspectral Unmixing With Spectral Variability Using Adaptive Bundles and Double Sparsity, IEEE TGRS", "Metric similarity regularizer to enhance pixel similarity performance for hyperspectral unmixing, Optik", "Hyperspectral Unmixing via Deep Convolutional Neural Networks, IEEE TGRS", "Fuzziness-based active learning framework to enhance hyperspectral image classification performance for discriminative and generative classifiers, plosOne", "Hyperspectral image classification: A benchmark", "Spatial Prior Fuzziness Pool-Based Interactive Classification of Hyperspectral Images, Remote Sensing", "Hyperspectral image classification using spectral-spatial LSTMs, Neurocomputing", and there are many more works exist.

Good luck!

Author Response

1. Extensions are not clearly explained! The problem statement is not clear and not well defined which lacks the novelty content. The Intuition regarding the pure spectral is obvious, lines 51-52. What's new in this? (The first comment of Expert reviewer 2)

Response: Thanks for expert reviewer’s comments. We clarify the extension on line 58-63 of the revised manuscript. We extended a lot of contents in the previous submitted paper compared with our previously published conference paper. The page number was extended to 14 pages from 7 pages in the conference papers. We added not only the ablation study for proposed method, the conventional sparse representation and the G-SOMP+ under different parameter but also conducted experiment on a real satellite image with much more bands: 128 compared with the used HS image datasets in the previous conference paper.

For lines 51-52 in the submitted manuscript in the first round, the intuition regarding the pure spectral is obvious. We agree with the expert reviewer. Based on this intuition, we formulate to calculating the spectral purity (mixing degree) using the surround pixel information in mathematical from, and adopt it to control the sparsity strength of the RGB vector for calculating the robust representation coefficients. This adaptive controlling of the sparsity strength is the main contribution in the HS image superresolution. We add the following sentence in the line 56-58 of the revised manuscript. As our best knowledge that it is the first time to adopting the spectral purity to adjust the sparsity strength of the representation coefficients for the HS image superresolution. Many Thanks.

2. here are many equations need to be revised, e.g. eq. 14.. (The second comment of Expert reviewer 2)

Response: Thanks for expert reviewer’s comments. We revised Eq. 14 and checked other Equation in the revised manuscript. Many thanks.

3. Extensive Englished editing required. There are many long sentences that are really hard to grasp, e.g. Line 33-38, 44-47. Many acronyms defined but never used e.g. Hyperspectral (HS) is defined however the full form is being used in several places. (The third comment of Expert reviewer 2)

Response: Thanks for pointing out this issue. We separated the long sentences into shorter sentences in the revised manuscript. We also gave the definition of acronym when the word is firstly appeared in the ‘Introduction’ section, and then use the acronym in the latter part. Thanks.

4. Experimental results are not very convincing (The comment of Expert reviewer 2)

Response: Thanks for expert reviewer’s comments. The mentioned work by the expert reviewer are mainly about the hyperspectral image classification. Our research goal is to estimate the HR-HS image from the available HR-RGB and LR-HS image, and is a HS image superreslution task. The popularly used datasets in HS image superresolution scenario are the CAVE and Harvard datasets. In order to conduct the fair comparison with the state-of-the-art methods (MF [27], CNMF [28], SNNMF [41], GSOMP [13], BSR [9]) for HS image superresolution, we evaluated our proposed method on the popularly used datasets: CAVE and Harvard). In addition, we also did experiments on a real satellite image with larger number of bands, and have proven the reasonable results using our method. It is possible to apply our method to any HS image but cannot provide the fair comparison with the state-of-the-art method. Thus we only manifested the HS image superresolution result with the Hyperspec-VNIR Chikusei image.Thanks.

Reviewer 3 Report

The authors propose a super-resolution method for hyperspectral images aided by HR RGB images Overall, the method is interesting and well explained.

The results are sufficient to characterize its performance. However, it is not clear why the authors did not test the "fixed" method on the Harvard dataset. I suggest to include it to have a broader assesment, unless there are specific reasons not to do it, in which case they should be reported in the paper.

The related work section should acknowledge the recent body of work on super-resolution techniques based on deep learning. Although to the best of my knowledge nobody worked on hyperspectral images, they represent the state-of-the-art for the super-resolution problem, outperforming methods based on dictionary learning and sparse representations. In particular, Ding Liu et al. "Non-Local Recurrent Network for Image Restoration", NeurIPS 2018 exploits non-local self-similarity for grayscale images. Andrea Bordone Molini et al. "DeepSUM: Deep neural network for Super-resolution of Unregistered Multitemporal images", arXiv:1907.06490, use deep learning to perform super-resolution from multitemporal single-band remote sensing images. Finally, Giuseppe Masi et al. "Pansharpening by convolutional neural networks", MDPI Remote Sensing use CNNs for pansharpening.

Author Response

1. it is not clear why the authors did not test the "fixed" method on the Harvard dataset. I suggest to include it to have a broader assesment, unless there are specific reasons not to do it, in which case they should be reported in the paper. (The first comment of Expert reviewer 3)

Response: Thanks for expert reviewer’s suggestions. We added the compared results with ‘Fixed’ method in Figure 4 and 5 of the revised manuscript.

2. The related work section should acknowledge the recent body of work on super-resolution techniques based on deep learning (The second comment of Expert reviewer 3).

Response: Thanks for pointing out this issue. We added the following work as the reference in the revised manuscrip. Many thanks.

[30] Ding Liu et al. "Non-Local Recurrent Network for Image Restoration", NeurIPS 2018

[31] Andrea Bordone Molini et al. "DeepSUM: Deep neural network for Super-resolution of Unregistered Multitemporal images", arXiv:1907.06490,

[25] Giuseppe Masi et al. "Pansharpening by convolutional neural networks".

Round 2

Reviewer 2 Report

The response to the fourth comment is not very convincing. HR-RGB images can also be generated from the LR-HS image cube and the proposed method can also be tested based on the quality. It will be a significant contribution. 

Author Response

Thanks for expert reviewer’s comments. We added two another satellite images: Salinas, and University of Pavia scenes (PaviaU), which are usually used in hyperspectral image classification problem, for evaluating our HS image super resolution method.

We consider Salinas and PaviaU data as the ground-truth HR-HS images, and simulate the corresponding LR-HS images via simply average down-sampling with factor: 6 for both horizontal and vertical directions, and the 8-band HR-MS images via generating the spectral response with Gaussian functions. Then we adopt different HS image superresolution methods on the simulated LR-HS and HR-MS images to estimate the HR-HS image for evaluation. The compared quantitative results are given in Table 3 and the recovered HR-HS images are shown in Figure 9. The explanation about the new used datasets are added in the revised manuscript (Red font).